# The Potential of Hydrogen Sulfide Donors in Treating Cardiovascular Diseases

**DOI:** 10.3390/ijms22042194

**Published:** 2021-02-23

**Authors:** Yi-Zhen Wang, Ebenezeri Erasto Ngowi, Di Wang, Hui-Wen Qi, Mi-Rong Jing, Yan-Xia Zhang, Chun-Bo Cai, Qing-Lin He, Saadullah Khattak, Nazeer Hussain Khan, Qi-Ying Jiang, Xin-Ying Ji, Dong-Dong Wu

**Affiliations:** 1Henan International Joint Laboratory for Nuclear Protein Regulation, School of Basic Medical Sciences, Henan University, Kaifeng 475004, China; yizhenwang713@163.com (Y.-Z.W.); ebenezerngowi92@gmail.com (E.E.N.); w15858109412@163.com (D.W.); qihuiwen0314@163.com (H.-W.Q.); j15184454140@163.com (M.-R.J.); 18834183556@163.com (Y.-X.Z.); ccb15345457076@163.com (C.-B.C.); hql1997010@163.com (Q.-L.H.); Saadullah271@gmail.com (S.K.); kakakhan3514@gmail.com (N.H.K.); 2Department of Biological Sciences, Faculty of Science, Dar es Salaam University College of Education, Dar es Salaam 2329, Tanzania; 3School of Nursing and Health, Henan University, Kaifeng 475004, China; 4Kaifeng Municipal Key Laboratory of Cell Signal Transduction, Henan Provincial Engineering Centre for Tumor Molecular Medicine, Henan University, Kaifeng 475004, China; 5School of Life Sciences, Henan University, Kaifeng 475004, China; 6Kaifeng Key Laboratory of Infection and Biological Safety, School of Basic Medical Sciences, Henan University, Kaifeng 475004, China; 7School of Stomatology, Henan University, Kaifeng 475004, China

**Keywords:** hydrogen sulfide, gas signaling molecule, cardiovascular homeostasis, heart function, cardiac protection

## Abstract

Hydrogen sulfide (H_2_S) has long been considered as a toxic gas, but as research progressed, the idea has been updated and it has now been shown to have potent protective effects at reasonable concentrations. H_2_S is an endogenous gas signaling molecule in mammals and is produced by specific enzymes in different cell types. An increasing number of studies indicate that H_2_S plays an important role in cardiovascular homeostasis, and in most cases, H_2_S has been reported to be downregulated in cardiovascular diseases (CVDs). Similarly, in preclinical studies, H_2_S has been shown to prevent CVDs and improve heart function after heart failure. Recently, many H_2_S donors have been synthesized and tested in cellular and animal models. Moreover, numerous molecular mechanisms have been proposed to demonstrate the effects of these donors. In this review, we will provide an update on the role of H_2_S in cardiovascular activities and its involvement in pathological states, with a special focus on the roles of exogenous H_2_S in cardiac protection.

## 1. Introduction

Cardiovascular diseases (CVDs) are non-communicable diseases targeting the heart (rheumatic heart disease, coronary heart disease, and congenital heart disease) and blood vessels (cardiovascular, peripheral arterial, deep vein thrombosis, and pulmonary embolism). According to the 2017 statistics, CVDs are the main cause of mortality worldwide, accounting for over 31% of all global deaths (55.98 million) and over 43% of all non-communicable disease-caused deaths (41.1 million) [1]. The risk factors for CVDs include smoking, diet, sedentary behavior, and pre-existing medical conditions such as obesity, diabetes, autoimmune diseases, and chronic kidney disease. Inflammation and oxidative stress have been identified as the main driving forces for atherosclerosis and, eventually, CVDs [2,3]. Currently, there are no specific cures for these diseases; however, several treatment options are used to improve the quality of life and survival of patients. In recent years, many studies have analyzed the possibility of using H_2_S in the treatment of several diseases including diabetes, chronic kidney diseases, and atherosclerosis [4,5,6]. The potential of H_2_S lies in its ability to regulate cellular mechanisms such as inflammation, apoptosis, and oxidative stress. In patients with CVDs, plasma free H_2_S levels are either elevated or decreased, depending on the disease type and severity [7,8]. Therefore, it is essential to determine the role of H_2_S in CVDs and potential mechanisms involved in H_2_S treatments.

H_2_S has been considered as a toxic/hazardous gas for over three centuries; this is mainly due to the discovery of several features including olfactory paralysis, sudden loss of consciousness, pulmonary edema, mucosal irritation, and keratoconjunctivitis in people exposed to high concentrations of H_2_S [9,10]. The gas has been reported to exhibit a steep exposure–response relationship [11]. However, it is important to note that prior to 1990, most of the studies involving H_2_S focused on its hazardous effects and very few addressed the positive role of the gas in the body. In the mid-1990s, essential evidence emerged linking the gas with the regulation of crucial body functions such as hippocampus activities and muscle relaxation [12,13]. Further studies identified the activation of adenosine triphosphate (ATP)-sensitive potassium (K_ATP_) channels and N-methyl-D-aspartate receptors as among the primary mechanisms used by H_2_S in modulating vascular functions and hippocampal long-term potentiation, respectively [14,15]. Together, these findings have opened up an exciting novel field in the regulation of human physiology by H_2_S.

Previous reviews have discussed the role of H_2_S in CVDs by addressing physiological processes including vascular muscle relaxation and autophagy [16,17]. However, limited information has been provided concerning the mechanisms involved; therefore, this review will illuminate recent studies demonstrating the roles of H_2_S in CVDs and pinpoint key mechanisms involved in the regulation.

## 2. Endogenous H_2_S Production

### 2.1. Enzymatic Production

In mammals, H_2_S production can occur both enzymatically and non-enzymatically. The enzymatic reactions that produce endogenous H_2_S are attributed to the presence of three enzymes, cystathionine beta-synthase (CBS), cystathionine gamma-lyase (CSE), and 3-mercaptopyruvate sulfur transferase (3-MPST) (Figure 1) [12,18,19]. CBS and CSE are localized in the cytosol, while 3-MPST is found in the mitochondria and cytoplasm. CSE and 3-MPST have been reported to be highly expressed in vascular tissues and associated with the regulation of smooth muscle relaxation [20,21].

CBS is a pyridoxal 5′phosphate (PLP)-dependent enzyme that is involved in the first step of the homocysteine (Hcy) pathway and is able to control the amount of methionine, leading to the sulfur transfer reaction [22]. CBS has three functional domains, a core domain, a heme-binding domain, and an S-adenosylmethionine regulatory site, with the former two serving as a site for PLP-catalyzed reactions and a redox sensor, respectively, whereas the latter helps in activating and stabilizing the enzyme through binding of the universal methyl donor S-adenosylmethionine to the regulatory domain [23,24,25]. CBS needs vitamin B6 as a coenzyme to catalyze the condensation of serine and Hcy to cystathionine [26]. The cystathionine is then converted into l-cysteine (Cys) and ketobutyric acid by CSE. CBS can also catalyze the condensation of Cys with Hcy to form l-Cys and H_2_S through a displacement reaction. It has been shown that the H_2_S produced by this reaction is at least 50 times more efficient than that produced by the hydrolysis of Cys in the β-elimination reaction [27]. The ratio of serine to l-Cys has been identified to be the primary determinant of the amount of H_2_S produced by CBS, with high l-Cys levels shown to favor the enzyme activities [23]. CBS is highly expressed in brain, liver, kidneys, and pancreas [28]. The main link between CBS and CVDs lies in their inverse correlation with Hcy levels, which is an independent factor for the progression of CVDs [29,30].

Similar to CBS, CSE produces H_2_S primarily by catalyzing Cys and Hcy [31]. There are many controversies in the scope of CSE expression. In several studies, CSE has been identified as the main H_2_S-producing enzyme in the vascular system [21,32]. Although, a study by Fu et al. could not detect CSE protein in mouse/rat cardiac tissue or cultured rat cardiac myocytes, despite its pharmacological inhibition resulting in over an 80% decline in H_2_S levels [33]. A recent report indicated that CSE is highly expressed in human aortic endothelial cells and is less expressed in coronary artery cells [34]. Besides, in high-fat-diet-treated human aortic endothelial cells, the enzyme is more robustly expressed as compared to controls [35]. It is worth noting that low intracellular calcium (Ca^2+^) levels can promote CSE-mediated H_2_S production, but the rise in Ca^2+^ levels in cells can inhibit the activity of CSE [36]. This effect has nothing to do with the activity of calmodulin, and the mechanism of regulation remains to be clarified.

Furthermore, 3-MPST is also strongly expressed in the vascular system and is localized in the vascular endothelium of the thoracic aorta, coronary aorta, and coronary artery [20,35,37]. It is also regarded as the main H_2_S-producing enzyme in the coronary artery. It produces H_2_S by using 3-mercaptopyruvic acid (3-MP) as a substrate in the presence of a reductant [20]. The production can occur in two ways: firstly, through the conversion of L-Cys to 3-MP by cysteine aminotransferase, followed by the capture of sulfur atom from 3-MP to form a persulfide and H_2_S in the presence of the reducing agent [20,38]. The second way involves a specific pathway that produces 3-MP and H_2_S from D-Cys catalyzed by D-amino acid oxidase [39]. The pathway is predominant in the cerebellum and kidney. It has been reported that the depletion of 3-MST levels has an age-dependent effect on the cardiovascular function of mice, with protective effects observed in young mice and deterioration of cardiac functions in adults [37]. Together, the above data suggest that 3-MPST has a crucial role in cardiac function; however, further studies are needed to examine the mechanism involved.

### 2.2. Non-Enzymatic Production

In mammalian cells, endogenous H_2_S is synthesized mainly by enzymatic reaction, although some H_2_S can be produced by non-enzymatic synthetic pathways. Non-enzymatic production of H_2_S involves Cys, thiosulfate, glucose, glutathione, sulfur atoms, and nicotinamide adenine dinucleotide and its reduced form [40,41,42].

## 3. Exogenous H_2_S

### 3.1. Direct Inhalation of Exogenous H_2_S Gas

Inhalation of H_2_S is the most direct method of administration. It has been shown that inhalation of 80 ppm of H_2_S at 27 or 35 °C for six hours could result in a reversible reduction in cardiovascular function and metabolic rate without affecting the blood pressure in mice [43]. Inhalation of H_2_S gas also reduces lipopolysaccharide (LPS)-induced acute lung injury by inflammatory responses in a mouse model [44]. Similarly, the inhalation of 40 or 80 ppm H_2_S attenuates the pro-inflammatory response induced by interleukin (IL)-1β, IL-6, and tumor necrosis factor-alpha (TNF-α) in rats after cardiac arrest and cardiopulmonary resuscitation [45]. However, treatment with high doses of 100–300 ppm H_2_S in sheep has been associated with the dose-dependent induction of adverse events such as hypoxemia, pulmonary vasoconstriction, and systemic vasodilation [46], indicating that high doses of the gas have detrimental effects. Regardless of the effects, direct inhalation of H_2_S faces many challenges such as flammability and toxicity, which may make safety management difficult.

### 3.2. Inorganic Sulfide

Inorganic sulfide donors that release H_2_S include sodium hydrosulfide (NaHS), calcium sulfide (CaS), and sodium sulfide (Na_2_S). NaHS and Na_2_S are the two most popular inorganic H_2_S donors in animal physiological studies. They have a very short half-life and can rapidly release the gas in a neutral aqueous environment [47]. Despite their popularity, several notable drawbacks have been reported, including the quick dissociation rate of these compounds, which could cause the actual concentration of H_2_S in the solution to be lower than expected. In addition, autoxidation and flammability also hinder their applicability in clinical settings. Compared to others, CaS is more stable and is considered as a most suitable inorganic H_2_S donor. It also serves as an active ingredient in the traditional medicine hepar sulfuris calcareum [48]. However, so far, there is no evidence available on the therapeutic potential of CaS in CVDs. In summary, although these sulfide salts, especially NaHS and Na_2_S, are in the mainstream of current research, their clinical application faces many challenges.

### 3.3. Natural H_2_S Donors

Garlic, rich in sulfur compounds, has been used by many researchers as a natural H_2_S releaser [49]. Through the catalytic action of alliinase enzyme, an alpha amino acid alliin which is a key component of garlic, is converted to allicin (diallyl thiosulfonate), which is then rapidly decomposed into diallyl sulfide (DAS), diallyl trisulfide (DATS), and diallyl disulfide (DADS) in aqueous solution [50]. These compounds can release H_2_S in human red blood cells by forming the key intermediate, namely hydrogen polysulfide (RSnH) [49]. In addition, sulfur compounds from garlic derivatives and aged garlic extract have been shown to play an important regulatory role in inducing cardioprotection and improving vessel function [51,52].

### 3.4. Synthetic H_2_S Donors

Synthetic H_2_S donors have many advantages, such as water solubility, stable storage, and harmless by-products. Some synthetic H_2_S donors also have specific and well-defined release mechanisms to help control the concentrations (i.e., they only release H_2_S in response to specific nucleophiles, light at specific wavelengths, or specific enzymes) [53]. Some of the common synthetic H_2_S donors include Lawesson’s reagent and its water-soluble derivative GYY4137, Glycine, N-(mercaptophenylphosphinyl)-, lithium salt (JK1), 2-dithiole-3-thiones (DTT), and DTT-nonsteroidal anti-inflammatory drugs (NSAIDs) [54]. GYY4137 has a dithiophosphate core structure and is currently the most well-known H_2_S slow-releasing agent. However, the compound has an obvious disadvantage—that is, it is usually prepared as a complex of methylene chloride, a compound which can be metabolized to generate formaldehyde and carbon monoxide (CO). Both products have been shown to participate in the regulation of cellular activities [55,56], raising the possibility of these compounds interfering with the effect of H_2_S. In recent years, many novel H_2_S donors have been discovered. For example, Park et al. reported a methodology to synthesize O-substituted phosphorodithioate-based H_2_S donors [57]. Similarly, Kang et al. have developed a series of thiophosphate sulfates called JK1, which are water-soluble compounds that show a slow, controlled, and pH-sensitive release of H_2_S in aqueous solutions [58]. In a phase 1 clinical trial, treatment with 20 mg/kg/day SG1002 (a H_2_S prodrug) has been reported to effectively ameliorate heart failure and improve heart function [59]. Another H_2_S donor, DTT, is commonly known to release the gas upon hydrolysis. DTT can also be partially attached to NSAIDs, resulting in conjugates such as NOSH aspirin, HS-aspirin (ACS14), HS-ibuprofen, HS-sulindac, S-sulindac, HS-naproxen, S-naproxen, and S-ibuprofen [60,61]. A DTT derivative of the triphenyl brominated tail, known as AP39, has also been synthesized. AP39 has mitochondrial targeting and antioxidant activities that could maintain cellular homeostasis by inhibiting the activation of the mitochondrial death pathway [62,63]. AP39 can not only significantly increase the viability of cardiomyocytes, but can also decrease apoptosis after cold hypoxia/reoxygenation, which has therapeutic potential in heart transplantation [62,63]. In addition, a novel donor NOSH-NBP has recently been synthesized from the mixture of NO and H_2_S, and the compound can significantly reduce the infarct volume and improve neurological deficits compared to NO-NBP or H_2_S-NBP in microglia cells and a mouse model [64].

## 4. Role of H_2_S in Vascular Function

### 4.1. Angiogenesis

Angiogenesis is the growth of new blood vessels from the existing capillaries or veins; it mainly includes the degradation of vascular basement membrane, activation, proliferation, and migration of vascular endothelial cells, and reconstruction and formation of new blood vessels and blood vessel networks. Angiogenesis is generally considered as an adaptive response to oxygen/nutrient deprivation engineered by vascular endothelial growth factor (VEGF) after ischemia or exercise. It has recently been found that sulfur amino acid restriction can regulate H_2_S through the general control nonderepressible 2/activating transcription factor 4 (ATF4) signaling pathway, thereby inhibiting mitochondrial electron transport and oxidative phosphorylation and ultimately promoting angiogenesis [65].

VEGFs are a family of proteins required for angiogenesis. They mainly include VEGF-A, VEGF-B, VEGF-C, VEGF-D, VEGF-E (virus encoding), and placental growth factor [66]. Hypoxia stimulates the secretion of VEGF and other angiogenic factors, leading to the formation of new blood vessels and the prevention of ischemic injury [67,68]. Such stimulation of VEGF and glycolytic genes (such as glucose transporter 1) by hypoxia is partially induced by the stabilization of transcription factors such as hypoxia-inducible factor 1-alpha (HIF-1α). A previous study shows that under hypoxia, peroxisome proliferator-activated receptor gamma coactivator 1-alpha can effectively regulate VEGF through the orphan nuclear receptor estrogen-related receptor-α pathway, thereby facilitating the formation of new blood vessels [69]. Similarly, mutant mice lacking CSE have been shown to exhibit a variety of pathological features, including slow angiogenesis and delayed wound healing [70]. Consistently, NaHS treatment has been reported to activate an endogenous angiogenic agent VEGF by elevating CSE levels, resulting in the inhibition of anti-angiogenic factors such as angiostatin, endostatin, and patatin and, subsequently, the formation of new blood vessels in the heart and reduction in the infarct area in myocardial infarction (MI) in mice [71].

Besides, a recent study demonstrated that suppression of CSE in human umbilical vein endothelial cells (HUVECs) results in mitochondria dysfunction by elevating the expression of soluble anti-angiogenic markers, fms-like tyrosine kinase-1 and endoglins, whereas supplementation of H_2_S by the donor AP39 could reverse the effect and improve cellular bioenergetics and reactive oxygen species (ROS) levels [72]. The treatment of HUVECs with a hybrid H_2_S and NO donor, ZYZ-803, has also been demonstrated to promote angiogenic properties by activating the signal of transducer and activator of transcription 3 (STAT-3)/Ca^2+^/calmodulin-dependent protein kinase II signaling cascade [73]. The above data confirm the role of H_2_S in angiogenesis and indicate that the CSE enzyme is the key factor in the regulation process.

Potassium (K^+^) channels and L-type Ca^2+^ channels are known to play important roles in CVDs [74]. It has been indicated that treatment with NaHS decreases intracellular concentration of Ca^2+^, thereby suppressing the activation of L-type Ca^2+^ channels and promoting the activation of the protein kinase C (PKC)/extracellular signal-regulated kinase (ERK) pathway, leading to the promotion of angiogenesis [75]. Likewise, a previous study showed that H_2_S induces angiogenesis by activating the K_ATP_ channels [70]. Treatment of chick chorioallantoic membrane with H_2_S-releasing micelles also promotes the formation of vessels, although the mechanism involved is yet to be explicated [76]. Similarly, Terzuoli et al. demonstrated that an angiotensin-converting enzyme (ACE) inhibitor, Zofenoprilat, induces the continuous production of H_2_S via the K_ATP_/protein kinase B (AKT)/endothelial nitric oxide synthase (eNOS)/ERK1/2 pathway to promote angiogenesis (Figure 2) [77]. Moreover, it has been revealed that H_2_S can effectively downregulate microRNA (miR)-640 through the VEGF receptor-2 (VEGFR2)/mammalian target of rapamycin (mTOR) pathway, hence increasing the level of HIF-1α and promoting angiogenesis [78]. Although H_2_S is a good angiogenic agent, due to its bell-shaped curve effect, it is important to be aware that high levels of CSE might pose a potential risk of plaque vulnerability in atherosclerotic plaques [79]. Altogether, the above evidence indicates that H_2_S can regulate angiogenesis through its involvement in the regulation of ion channels, miRNA, and other angiogenic components, although higher levels of CSE might also promote plaque formation.

Nitric oxide (NO) also plays a vital role in the formation of blood vessels and in the progression of CVDs [80]. A previous study reported that NaHS can improve the viability and function of chronic ischemic tissue, resulting in increased angiogenesis by regulating VEGF and HIF-1α in a NO-dependent manner under hypoxic conditions [81]. H_2_S can also activate the NO/cyclic guanosine 5′-monophosphate (cGMP) and the phosphoinositide 3 kinases (PI3K)/AKT pathways, leading to the downstream regulation of vascular angiogenesis [82]. Furthermore, the silencing of CSE or CBS in larval zebrafish has been shown to induce brain vascular developmental defects, while treatment with GYY4137 could reverse the changes by promoting the production of NO [83]. In summary, these data imply that H_2_S donors can increase vascular angiogenesis via the H_2_S/NO axis and confirm its potential for utilization in the treatment of CVDs.

### 4.2. Migration

Endothelial cell migration is a key component of angiogenesis and critical to wound healing. Under both hypoxic and normoxic conditions, inhibition of CSE and CBS could result in the promotion and suppression of the migration of HUVECs, respectively, whereas the suppression of 3-MPST reduces the migration only in the hypoxic condition, suggesting that these enzymes are essential in the regulation of migration [84]. It is speculated that the difference might exist because endogenous H_2_S produced by CBS, CSE, and 3-MPST may have different target molecules. Furthermore, a recent study shows that inhibition of 3-MPST significantly reduces endothelial cell proliferation, migration, and tubular network formation by regulating VEGF [85]. This metabolomic analysis also reveals that 3-MPST attenuation mitigates mitochondrial respiration and ATP production but increases glucose uptake. Besides, another study reports that exogenous H_2_S improves the migration of high-glucose-treated HUVECs by suppressing the expression of DNA methyltransferase 1 protein, resulting in the subsequent elevation of miR-126-3p [86]. Moreover, recent studies indicate that treatment with GYY4137 suppresses the migration of HUVECs by increasing protein S-sulfuration [87] and activating the PI3K/AKT pathway [88]. Although the changes in the effects have been suggested to be due to the nature of the donor as compared to NaHS, which has been mostly used in the other studies, the area needs further exploration. In summary, the above evidence suggests that H_2_S-producing enzymes are crucial for regulation of the migration of endothelial cells and H_2_S supplementation could improve the activity.

### 4.3. Vasodilation

Vasodilation is a vital process involved in the regulation of blood pressure, and its dysregulation is considered as a risk factor for CVDs [89]. NO, angiotensin II (Ang II), K^+^, and Ca^2+^ channels play crucial roles in the regulation of blood flow and vasodilation [14,90,91]. Reduction in endogenous H_2_S-induced vasodilation has been reported in hypertension patients as compared to adults [92]. In mesenteric stem cells, downregulation of CBS reduces the production of H_2_S and dilation of mesenteric arteries [93]. Furthermore, treatment with the CSE inhibitor DL-propargylglycine (PAG) enhances vascular resistance and increases bloop pressure in rats, indicating that H_2_S is involved in the regulation of vasodilation [94]. Correspondingly, stimulation of H_2_S production with PLP enzyme significantly elevates H_2_S levels and improves vascular relaxation together with oxidative and nitrosative statuses in adult rats [95]. Similarly, treatment with NaHS has also been shown to induce the relaxation of human uterine artery, partly by triggering the large conductance calcium-activated and voltage-dependent K^+^ (BK) channels [96]. An isothiocyanate vasodilating compound, sulforaphane, has also been reported to induce its effect by activating K_ATP_ and BK channels in a CBS/CSE/H_2_S-dependent manner [97]. In addition, H_2_S also induces vasodilation by promoting NO release and ultimately activating the NO/cGMP/soluble guanylyl cyclase/protein kinase G pathway in pial arteries [98]. A previous study further revealed that treatment with Na_2_S improves the vasodilation of endothelial cells by activating transient receptor potential cation channel V-4 [99]. Otherwise, a glucagon-like peptide 1 inhibitor exenatide has been identified to facilitate vasodilation and decrease aortic blood pressure by stimulating H_2_S, NO, and CO production, consequently activating the voltage-dependent K^+^ channel subfamily KQT member 5 type pathways [100]. However, compared to other gasotransmitters, H_2_S can induce a stronger effect. Collectively, these data signify that H_2_S treatment improves vascular function, including vasodilation, by regulating ion channels and multiple signaling pathways.

### 4.4. Apoptosis

Apoptosis is one of the fundamental processes involved in the progression of CVDs as it participates in the loss of myocytes. High apoptotic rate is associated with poor prognosis for patients with heart failure [101]. A recent study demonstrated that downregulation of the CSE/H_2_S system exacerbates hypoxia-induced apoptosis in H9c2 cardiomyoblast cells and mesenchymal stem cells, while supplementation with the H_2_S-releasing drug trimetazidine reverses the effects in H9c2 cells [102]. Similarly, exogenous H_2_S can effectively mitigate hydrogen peroxide (H_2_O_2_)-induced apoptosis by improving beclin-2 (Bcl-2)/Bcl-2-associated X protein (Bax) status in cardiomyoblasts [103]. In smoke-induced left ventricular dysfunction rat models, treatment with NaHS suppresses the elevated pro-apoptotic activities by activating the PI3K/AKT and deactivating the c-jun N-terminal kinase (JNK) and p38 mitogen-activated protein kinase (MAPK) pathways [104]. In addition, it has been shown that treatment with GYY4137 can markedly prevent glucose-induced damage in H9c2 cells by reducing apoptosis via the inhibition of the STAT-3/HIF-α signaling cascade [105]. Collectively, the above evidence indicates that apoptosis-induced progression of myocardial dysfunction can be attenuated with the elevation of H_2_S levels.

### 4.5. Oxidative Stress

Oxidative stress is a common pathological process characterized by the imbalance between the oxidants and antioxidants status of a cell which, results in cellular damage. A previous study showed that oxidative stress is significantly correlated with the severity of acute coronary syndrome in patients [106]. Oxidative stress is also linked with oxidation-induced modifications of low-density lipoproteins (oxLDL), which has a strong association with the progression of CVDs [107,108]. In healthy individuals, fasting blood levels of H_2_S have been confirmed to be negatively correlated with the ratio of LDL/high-density lipoproteins, which is also a risk factor for CVDs [109]. A previous study reported that NaHS treatment can prevent hemin-induced oxLDL elevation and oxidative stress in smooth muscle cells (SMCs) through the elevation of H_2_S levels [110]. Similarly, pre-treatment with NaHS has been shown to prevent neonatal mouse cardiomyocyte damage induced by H_2_O_2_ through the alleviation of LDL-induced ROS increase and upregulation of sirtuin (SIRT)-1 expression [111]. Besides, treatment of H9c2 with GYY4137 also suppresses glucose-induced oxidative stress by decreasing the levels of malondialdehyde, ROS, and NADPH oxidase 2, as well as increasing that of superoxide dismutase (SOD) through STAT-3/HIF-α inactivation [105]. Strangely, under hypoxic conditions, the elevation of HIF-α has been reported following supplementation of H_2_S using the fast-releasing donor NaHS [78,81]; however, the slow-releasing donor GYY4137 exhibits the opposite effect, suggesting that the nature of the donor might influence the downstream effect on HIF-α. Therefore, further studies are required to address the event. Besides, a previous study also showed that treatment with S-propargyl-cysteine (SPRC) improves the antioxidant status in glucose-induced oxidative stress in H9c2 by promoting Nfr-2 nuclear translocation via AKT and CSE/H_2_S pathways [112]. The information above implies that H_2_S donors can be used to treat CVDs by targeting oxidative stress.

### 4.6. Autophagy

Autophagy is one of the main regulators of cell disturbances, especially oxidative stress-induced cellular defects. It involves self-digestion of the cell and recycling of the vital nutrients. Evidence shows that autophagy is explicitly aggravated in end-stage heart failure patients, and it leads to secondary cell death occurring via oncosis and necroptosis [113]. Since moderate activation of autophagy has shown cytoprotective effects [114], it is possible that its hyperactivation results in detrimental effects to cardiac cells. Besides, a recent study demonstrated that treatment of human cardio fibroblast cells with NaHS can effectively attenuate autophagy activation and prevent oxidative stress and associated mitochondria dysfunctions induced by H_2_O_2_ [115]. The anti-autophagic effect of NaHS is reported to be mediated via the activation of the PI3K/serum and glucocorticoid-regulated kinase 1/glycogen synthase kinase-3 beta (GSK-3β) signaling pathway [116]. Similarly, a previous study indicated that NaHS treatment can suppress apoptosis and autophagy in smoking-induced left ventricular dysfunction in rats via inhibition of the p38 MAPK and JNK pathways, as well as activation of the PI3K/AKT and AMP-activated protein kinase (AMPK)/mTOR pathways [104]. Protective autophagy has also been reported to be induced by GYY4137 through the elevation of SIRT-1 levels and, subsequently, nuclear translocation and deacetylation of the Forkhead box protein O1 (FOXO1) signaling cascade in HUVECs [117]. In summary, the above information implies that H_2_S donors can potentially suppress exacerbated autophagic cell death and maintain protective autophagy by regulating the involved pathways.

### 4.7. Inflammation

Inflammation is one of the main driving forces for CVDs. In patients with systemic lupus erythematosus, inflammation has been identified as a common risk factor for CVDs [118]. High levels of inflammatory markers such as C-reactive protein, TNF-α, and IL-6 have been associated with an increased risk for the development of CVDs [119,120,121]. Inflammation and oxidative stress are usually intertwined and they can regulate one another. In cardiac cells, the levels of TNF-α have been positively linked with the production of ROS [122], and the oxidative stress-induced oxLDL modifications with inflammatory responses [123]. It has been shown that treatment with 100 µM NaHS effectively reduces pro-inflammatory activities by S-sulfurating Cys-269 of the c-jun/activator protein-1 (AP-1) and subsequently activating p62 and SIRT-3 cascades in macrophages [124]. A recent study also revealed that NaHS treatment can effectively reduce smoking-induced inflammation in alveolar epithelial cells by mitigating the JNK, ERK, and prolyl hydroxylase 2/HIF-α/MAPK pathways [125]. Otherwise, pre-treatment with NaHS prevents doxorubicin-induced elevation of IL-6, IL-1β, and TNF-α expression in H9c2 cells by deactivating the p38/nuclear factor kappa B (NF-қB) pathway [126]. Similar effects have been observed following treatment with NaHS in coxsackie virus B3-infected rat cardiomyocytes [127]. Moreover, it has been shown that treatment with SPRC inhibits intercellular adhesive molecule-1 and TNF-α elevation induced by lipopolysaccharides in H9c2 cells through the suppression of NF-қB and activation of the PI3K/AKT pathway [128]. Interestingly, treatment with NaHS has been associated with increased activities of c-jun/AP-1 via sulfuration in macrophages [124] and inhibition of the JNK pathway in cardiomyocytes and epithelial cells A549 [104,125]. With different factors such as cell type, duration, and drug concentration known to influence the effects of the donors, further studies are needed to determine the exact reason for the disparity. Altogether, the above data suggest that H_2_S donors can regulate inflammatory pathways and serve as potential treatment options for CVDs.

## 5. H_2_S and CVDs

### 5.1. Hypertension

Hypertension is a key risk factor for CVDs and directly or indirectly contributes to at least nine million deaths each year. The World Health Organization has identified hypertension as one of the most important risk factors for morbidity and mortality in the world. Currently, about one-third of patients with hypertension have not been diagnosed, and about half of those diagnosed have not taken blood pressure drugs [129]. In hypertensive children, the levels of H_2_S are significantly reduced as compared to healthy subjects [130]. Moreover, the H_2_S level in hypertensive patients can be correlated with the severity of the disease [131]. It has been found that the levels of H_2_S in serum, heart, and aorta of CSE knockout mice are significantly reduced, and CSE-deficient mice show hypertensive features and reduced endothelium-dependent vasodilation [21]. A recent study also reported that the deletion of 3-MPST induces a dual age-dependent effect, where in younger mice, it promotes cardioprotection, while in adults, it is associated with the progression of hypertension [38]. Alternatively, treatment with NaHS (5.6 mg/kg/day, intraperitoneal) reduces the incidence of hypertension by inhibiting the release of renin from glomerular cells and Ang II levels through the suppression of cyclic adenosine monophosphate [132]. In addition, another study reports that NaHS treatment can reduce high salt intake-induced hypertension in Dahl rats and restore structural remodeling of the thoracic aorta [133]. Ang II is a vital component in the regulation of vasodilation. It has been revealed that exogenous NaHS can reduce blood pressure, endothelial dysfunction, and vascular oxidative stress in chronic Ang II-induced hypertension in mice [134]. Further evidence shows that H_2_S exerts a protective effect by interfering with the zinc (Zn^2+^) activity of the ACE’s active center in HUVECs, thereby reducing the production of Ang II, inhibiting the degradation of bradykinin, and ultimately inducing the vasodilatation of blood vessels [135]. The CSE/H_2_S axis has also been shown to reverse Ang II-induced hypertensive features in lymphocytes and a mouse model partially by sulfurating liver kinase B1, activating AMPK pathway, and promoting the differentiation of regulatory T cells [136].

Moreover, several studies have shown that treatment with H_2_S donors can facilitate vasodilation by elevating NO bioavailability and modulating the renin–angiotensin system in hypertensive rats [134,137]. The protective effect of H_2_S on the endothelium has been shown to be mediated via the downregulation of bone morphogenetic protein 4/cycloxygenase-2 signaling pathways in hypertensive rats [138]. A recent study also indicated that H_2_S attenuates hypertension-induced endothelial dysfunction in the renal artery by activating the peroxisome proliferator-activated receptor (PPAR)/PI3K/AKT/eNOS or PPAR/AMPK/eNOS pathways [139]. In summary, the data above indicate that H_2_S donors can be used to regulate the production and activities of Ang-II and NO, two of the highly essential components associated with high blood pressure that are commonly dysregulated in hypertension. In addition, H_2_S can interact with several signaling pathways such as AMPK and PI3K/AKT to reduce endothelial dysfunctions caused by hypertension. Future studies should address the toxicities, clearance mechanisms, and side effects associated with these donors.

### 5.2. Cardiac Fibrosis (CF)

CF is a global health problem characterized by increased stiffness in the left ventricle and resistance to heart contraction and relaxation. CF occurs in almost all types of cardiac diseases, including MI, aortic stenosis, diabetic cardiomyopathy, and hypertrophic cardiomyopathy. This adverse change eventually leads to heart failure. Cardiac fibroblasts and cardiac myocytes are the two most essential cell types in the heart, and the former are responsible for extracellular matrix (ECM) homeostasis [140]. During ventricular remodeling after injury, the area between muscle cells and blood vessels becomes smaller due to the deposition of collagen and growth factors resulting from repeated changes in the hemodynamic load [141]. CF caused by hypertension, aortic stenosis, and other diseases can induce more recessive mesenchymal and perivascular collagen deposition, whereas the one caused by valve regurgitation is characterized by a large amount of non-collagenous matrix [142]. CF can be divided into two types: interstitial fibrosis and replacement fibrosis. In interstitial fibrosis, the non-conductive fibrous collagen network between myocardial slices may promote ventricular tachycardia by inducing focal ectopic activity and by slowing or blocking conduction [143]. Fibrosis is basically a compensatory mechanism used to maintain the structure of the heart, triggering “repair” fibrosis by activating phenotypic transformed fibroblast-like cells. The structural integrity of the myocardium is preserved through this scar tissue; however, the structure might also increase the propensity for tissue stiffness and arrhythmia [144]. The population of activated myofibroblasts continues to generate Ang II and fibrotic growth factors (e.g., transforming growth factor-l), which act as the signal transductor–effector signaling pathways for type I collagen synthesis and participate in fibrosis.

With respect to H_2_S, it has been demonstrated that NaHS treatment can reduce the formation of cardiac fibroblasts by preventing the activation of K^+^ channels [145]. In addition, a previous study reports that NaHS can effectively reduce CF by inhibiting the expression of Ang II-induced motilin, connective tissue growth factor and type I collagen, as well as the expression of heme oxygenase-1 in cardiac fibroblasts, a process associated with the reduction of the NOX4-ROS-ERK1/2 signal transduction axis [146]. Recent studies further reveal that treatment with NaHS can markedly improve CF in diabetic rats by promoting CSE synthesis and autophagy through stimulation of the SIRT-6/AMPK and PI3K/AKT/eNOS pathways [147,148]. Similarly, in a thyroxine-induced CF rat model, H_2_S induces a protective effect by regulating autophagy via activation of the PI3K/AKT pathway [149]. Moreover, another study suggests that H_2_S can protect against endoplasmic reticulum (ER) stress-induced endothelial-cell-to-mesenchymal transformation through attenuation of the Src pathway, thus mitigating CF [150]. However, more studies are needed to clarify the mechanism. Besides, it has also been reported that daily administration of GYY4137 for 4 weeks can significantly reduce the degree of myocardial fibrosis and systolic blood pressure in spontaneously hypertensive rats by decreasing the total collagen contents of myocardium, *α*-smooth muscle actin levels, and oxidative stress through the deactivation of the transforming growth factor beta (TGF-β)/Smad2 pathway [151]. Further reports show that H_2_S can attenuate matrix deposition and myocardial fibrosis in diabetic rats and improve the matrix metalloproteinase (MMP)/tissue inhibitor of metalloproteinase (TIMP) status through the inhibition of TGF-β-mediated collagen accumulation together with Janus kinase (JAK) 1/2-signal transducer and activator of transcription (STAT) 1/3/5/6 and wingless and Int-1 (Wnt) signaling cascades, thereby resulting in a subsequent decrease in type I and type III collagen and CF [152,153]. Besides, another study indicates that exogenous H_2_S can also improve myocardial fibrosis in mice with diabetic heart disease by triggering FOXO-1 phosphorylation and nuclear rejection [154].

Besides, miRNAs play a crucial role in the regulation of various cellular activities; hence, their dysregulation has detrimental effects and can lead to the progression of disease states. It has been shown that pathological remodeling responses induced by acute MI are accompanied by characteristic changes in the expression of specific miRNA sets [155]. Shen et al. demonstrate that the overexpression of miR-30 effectively downregulates H_2_S levels by directly targeting CSE enzyme, thereby promoting cardiac insults; however, the event can be rescued by H_2_S supplementation with SPRC [156]. Further study indicated that the administration of NaHS reduces autophagy and matrix deposition in mice by attenuating the expression of miR-21, miR-34a, miR-214, and miR-221 as well as activating the PI3K/AKT signaling pathway, resulting in the alleviation of CF [149,157]. Together, these data confirm the potential of H_2_S in targeting miRNAs and other dysregulated parameters involved in the development of CF. Since CF is a repair mechanism for damaged cells, further studies are needed to check the effects of H_2_S on different stages of CF and determine whether supplementation with donors in the initial stages can affect the mechanism and promote cell damage.

### 5.3. Cardiac Hypertrophy (CH)

CH is primarily an adaptive response to restore cardiac function by increasing contraction rate and the thickness of the left ventricular wall in response to stimuli [158]. The condition has been identified as a risk factor for congenital heart failure and mortality. CH can be divided into physiologic and pathologic CH. Physiological CH is a compensatory mechanism for the body to cope with adverse situations, with the purpose of stabilizing cardiac output. The condition is most common in children, pregnant women, and athletes. Alternatively, pathological CH is common in patients with CVDs such as hypertension and MI [159]. Cellular processes such as apoptosis, oxidative stress, and inflammation are highly correlated with the progression of the condition [160]. It has been reported that treatment with NaHS improves heart function by suppressing hypertrophy via upregulation of miR-133a and inhibition of intracellular ROS [161]. Similarly, a recent study indicated that H_2_S-induced elevation of miR-133a results in the downstream regulation of the Ca^2+^/calcineurin/nuclear factors of activated T-cells pathway [162]. Furthermore, the treatment of a CH-induced rat model with NaHS markedly restores ventricular function by reducing the expression of connexin-43 and activating K_ATP_ channels [163]. A recent study further shows that treatment with SG1002 can prevent Hcy-induced hypertrophy in CBS^+/-^ mice and improve cardiac function by reducing MMP activity and TGF-β activation [164]. Another H_2_S-releasing compound, allyl methyl sulfide, has been shown to attenuate isoproterenol-induced CH in rats by restoring the levels of MMP-2 and -9 as well as diminishing ROS and apoptosis activities [165]. In addition, a previous study showed that NaHS treatment can efficiently reduce the oxidative stress, apoptosis, and cardiac dysfunction in isoproterenol-induced CH by decreasing the expression of NOX4, caspase-3, and cytochrome c, which are the key players in the cardiac mitochondrial pathway [166]. Furthermore, the anti-hypertrophy effect of H_2_S can also be mediated by antagonizing Ang II and norepinephrine levels, as well as reducing the degree of oxidative stress and improving the pulse rate [167,168].

Besides, it has been revealed that treatment with Na_2_S enhances proteasomal activities and reduces the accumulation of damaged proteins by regulating Nrf-2 signaling in a wild-type mouse model of CH; however, the treatment could not show any significant protective effect in Nrf-2 KO mice [169]. A recent study also revealed that GYY4137 treatment can regulate the transcriptional activity of KLF-5 by mediating the S-sulfuration of specificity protein-1 at Cys-664, thereby preventing CH [170]. Moreover, H_2_S treatment enhances the expression and activity of SIRT-3, thereby improving mitochondrial function, and reduces apoptosis and oxidative stress activities in CH mouse models [171,172]. Overall, the above evidence confirms the potential of H_2_S in the suppression of CH by regulating mitochondrial pathways, oxidative stress, and inflammation-related pathways; however, the effects of these drugs on toxicity in normal cells and surrounding tissues need to be further investigated.

### 5.4. Cardiac Valve Calcification (CVC)

Cardiac valve calcification (CVC) is a common condition associated with chronic kidney diseases and elderly patients. It usually affects the aortic and mitral valves and may lead to the dysfunction of heart valves and stenosis. CVC is associated with an increased risk of cardiovascular morbidity and mortality [173]. In the current guidelines of the European society of cardiology and cardiology surgery of 2017, surgical treatment is the gold standard for valvular vein diseases, but safe and effective treatment options are currently lacking for elderly patients due to high surgical risk. Calcification is characterized by intracellular or extracellular deposition of minerals, mainly hydroxyapatite. A recent study indicated that H_2_S donors (NaHS, Na_2_S, GYY4137, AP67, and AP72) can significantly inhibit osteoblasts’ transdifferentiation in a dose-dependent manner, while downregulation of CSE or CBS exacerbates the condition, suggesting that the endogenous CSE/CBS-H_2_S system is involved in the calcification of heart valves [174]. Similarly, another study showed that treatment with AP72 can efficiently prevent inflammatory activities and calcification of the aortic valves of apolipoprotein E knocked-out mice (ApoE^-/-^) through deactivation of the NF-қB/runt-related transcription factor-2 pathway, whereas the silencing of CSE/CBS exacerbates the condition [175]. Currently, studies on H_2_S and cardiac valve calcification are still in their infancy; hence, more studies are needed to demonstrate the effect of H_2_S on CVC in a three-dimensional manner by illuminating more on the mechanisms involved, downstream effects, and toxicity.

### 5.5. Takotsubo Cardiomyopathy (TCM)

Takotsubo cardiomyopathy (TCM), also known as stress-induced cardiomyopathy, is a condition caused by stressful events that can lead to the development of cardiovascular dysfunction symptoms and heart failure [176]. In the United States, there has been a threefold increase in the prevalence of TCM from 2007 to 2012, with one study showing that the condition is more common in women, but the mortality rate is much higher in men [177]. Inflammation and oxidative stress have been identified as vital processes which are commonly elevated in TCM patients and animal models [178,179]. Plasma H_2_S levels and cardiac expressions of CSE and 3-MPST enzymes are reduced in a rat TCM model [180]. Besides, overactivation of β-adrenergic signaling is positively associated with the development of TCM; hence, treatment with β-adrenergic agonist (isoprenaline) has been widely used to induce the disease in animal models [181]. In addition, treatment with NaHS has been demonstrated to successfully suppress cardiac defects in isoprenaline-induced TCM rats by reducing ROS through the downregulation of NADPH oxidase [180]. However, the mechanism involved in H_2_S-mediated regulation of TCM is yet to be revealed. In brief, the above data indicate that H_2_S donors have potential in treating TCM; however, more studies are needed to determine the mechanisms, pathways, and downstream effects associated with the treatments.

### 5.6. Diabetic Cardiomyopathy (DCM)

Diabetic cardiomyopathy (DCM) is basically a diabetes-associated condition characterized by high oxidative stress, mitochondrial defects, and, ultimately, myocardial dysfunctions [182,183]. It is the leading cause of death in diabetic individuals. Numerous studies have reported the protective effects of H_2_S donors in the attenuation of DCM through the regulation of inflammation, oxidative stress, and apoptosis activities [184,185,186]. NaHS treatment could improve cardiac function by targeting ROS and autophagic activities in a mouse DCM model through the elevation of Kelch-like ECH associated protein 1 (Keap-1) expression [187]. Besides, it has been shown that treadmill exercise can stabilize the decreased levels of CSE and CBS in high-fat-fed mice, thereby elevating H_2_S levels and subsequently preventing the development of DCM [188]. Pre-treatment of H9c2 cells, neonatal rat cardiomyocytes, and a mouse model with GYY4137 significantly mitigates high-glucose-induced oxidative stress and apoptosis through the promotion of FOXO-1 phosphorylation and nuclear translocation, indicating that the compound can potentially prevent the progression of DCM [154]. Similarly, NaHS treatment can effectively suppress cardiac dysfunction by preventing the ubiquitination of myosin heavy chain 6 and myosin light chain 2 and their corresponding interaction with muscle ring finger-1 (an E3 ubiquitin ligase) by sulfurating the ligase at Cys-44 [189]. In the cardiomyocytes of high-glucose-fed rats, NaHS treatment protects cardiac function by preventing ROS accumulation and apoptosis via inhibition of the Wnt/β-catenin pathway [190]. In summary, the above data reveal that H_2_S donors can effectively prevent the progression of DCM by targeting oxidative stress, apoptosis, autophagy, and inflammatory processes via their respective pathways. With the available information, further studies can focus more on the proteins regulated, their interactions, and possible side effects of these drugs.

### 5.7. Atherosclerosis

Atherosclerosis is the leading cause of CVDs-associated deaths. It is characterized by chronic inflammation of the blood vessels as a result of the deposition of lipids on the arterial walls [191,192]. Estrogen deficiency is one of the risk factors for menopausal atherosclerosis in women and its supplementation has been shown to reduce the risk in animal models [193]. With respect to H_2_S, evidence shows that female CSE^-/-^ knockout atherosclerotic mice have reduced estrogen levels and more plaque lesions than their wild type counterparts, suggesting an interaction between estrogen and CSE in the development of the disease [194]. It has been demonstrated that hyperhomocysteinaemia induces CSE nitrosylation at Cys-252, -225, -307, and -310 in atherosclerotic mouse models, while treatment with H_2_S donors induces anti-atherosclerotic properties by triggering the sulfidation of CSE at the same Cys residues [195]. Similarly, previous studies have shown that H_2_S treatment can attenuate the development and progression of atherosclerosis by facilitating NO production, protein sulfidation, and S-nitrosylation [196,197,198]. It has also been reported that treatment of atherosclerotic mice with NaHS and GYY4137 can prevent plaques, macrophage infiltration, and can reduce serum homocysteine levels through the alleviation of caspase-3 and -9 and MMP-9 expression via the downregulation of ERK/JNK signaling [199]. In addition, the donors induce the activation of SIRT-1 by mediating its sulfidation at its two Zn^2+^ finger domains, resulting in the reduction in its ubiquitination and subsequent degradation [200]. Inhibition of the PI3K/Akt/eNOS signaling pathway in glucose-injured HUVECs and the associated elevation of ROS and apoptotic activities can be reversed with pre-treatment with 400 μM NaHS for 30 min [201]. Another study reports that treatment with alpha-lipoic acid or NaHS can attenuate diabetes-induced vascular dysfunctions in SMC in rats by elevating H_2_S levels and reducing autophagy by targeting the AMPK/mTOR pathway [202]. Similarly, NaHS treatment prevents the development of atherosclerosis in uremic-accelerated atherosclerotic ApoE^-/-^ mice through the suppression of TGF-β/Smad3 and activation of the conventional PKC βII/AKT/eNOS pathway [203,204]. A recent study also showed that treatment of diabetes-stimulated atherosclerotic cells and mice with GYY4137 significantly reduces the elevation of pro-inflammatory activities by promoting the activation of PI3K/AKT and the deactivation of NLRP3 and Toll-like receptor-4 signaling [205,206]. Correspondingly, it has also been revealed that treatment of atherosclerotic swine with sodium thiosulfate can markedly reduce hemorrhagic shock-induced lung impairment during resuscitation by regulating glucocorticoid receptors [207]. Moreover, Lin et al. report that NaHS can induce anti-atherosclerotic effects by elevating the levels of Ang II converting enzymes, thereby facilitating blood flow and suppressing pro-inflammatory activities [208]. Several key mechanisms and pathways involved in atherosclerosis have been reported to be reversed with H_2_S supplementation; however, no information is available on the clearance mechanism of these donors nor on the side effects on body organs such as the liver and kidneys, which are commonly known to be induced by drugs. Thus, further studies should be conducted in this field.

### 5.8. Myocardial Ischemia/Reperfusion (M-I/R) Injury

Myocardial ischemia refers to the reduction in blood flow from the coronary artery, which can result in the reduction in the supply of oxygen and nutrients to the heart muscles. On the other hand, reperfusion is a vital process initiated to restore the blood flow; however, its delay can lead to myocardial injury and cells’ death that are partially injured [209]. Dysregulation in antioxidant status, apoptosis, inflammation, pH, and intracellular Ca^2+^ are the key features characterizing lethal reperfusion injury [210,211]. A previous study showed that treatment of M-I/R rats with the CSE inhibitor PAG significantly increased infarct size, suggesting that suppression of the gene aggravates the condition [212]. In a recent study, endothelia cell-specific CSE overexpressed transgenic mice were also revealed to display high protective effects induced through the activation of eNOS and elevation of NO production when subjected to M-I/R [213]. In addition, CSE knockout mice exhibit high endothelial dysfunction but without aggravating the induced M-I/R. Similarly, another recent study indicated that the deletion of CSE in mice is associated with the increase in mean arterial pressure and miR21a expression compared to a control group; however, no intensification of M-I/R effects could be observed [214]. These findings suggest that CSE/H_2_S play a crucial role in cardioprotection, although the suppression of CSE alone cannot promote the progression of M-I/R. Otherwise, several studies have shown that the elevation of H_2_S levels can essentially prevent the progression of M-I/R injury by regulating oxidative stress, apoptosis, and inflammation processes [55,215,216,217]. The encapsulation of DATS with mesoporous iron oxide nanoparticles has also been reported to improve the efficiency of the donor by increasing the biocompatibility, specificity, and prolonging the drug release rate as well as reducing side effects [218]. A previous study suggests that treatment of M-I/R diabetic mice with Na_2_S effectively attenuates the progression of injury by reducing ROS levels and caspase-3 expression by activating the ERK1/2-reperfusion injury salvage kinase pathway [219]. Treatment with GYY4137 has also been shown to mitigate M-I/R injury in diabetic mice by activating the pleckstrin homology domain leucine-rich repeat protein phosphatase-1/AKT/Nfr-2 pathway [220], whereas NaHS utilizes the CSE/NO axis to induce a protective effect [221]. In type 1 diabetic mice, treatment with DATS promotes the activation of the SIRT-1 and Nfr-2 antioxidant pathway together with the deactivation of the protein kinase RNA-like endoplasmic reticulum kinase-eukaryotic initiation factor 2 alpha/ATF4/C-EBP homologous protein signaling pathway, thus reducing ER stress activities [222]. In another study, the H_2_S donor ADT has been shown to restore autophagy flux through activation of the AMPK pathway, thereby diminishing M-I/R injury [223]. A recent study also shows that treatment of a rat model with NaHS can effectively abolish M-I/R injury-induced elevation of oxidative stress and apoptosis by inhibiting activation of the JNK pathway [224]. Moreover, the long non-coding RNA Oprm1 has also been reported to suppress M-I/R injury by elevating the levels of H_2_S, activating the PI3K/AKT pathway, and reducing the activity of HIF-α [225]. Another study reports that treatment of M-I/R-injured rats with NaHS can potentially reduce pro-apoptotic activities as evidenced by the elevation of Bcl-2 levels and downregulation of Bax induced via the inhibition of GSK-3β/β-catenin signaling cascades [226]. Collectively, the above data confirm the role of H_2_S in the progression of M-I/R injury and indicate that the supplementation of H_2_S donors is a promising therapeutic option due to the ability to target numerous pathways that are dysregulated in the condition. Since most of the mechanisms and pathways have been studied, future studies can focus on the toxicity of these donors in body organs, as well as the rate and mechanisms of clearance.

## 6. Conclusions

In this review, we summarize the mechanisms of H_2_S donors in CVDs treatment by using the latest knowledge available with the aim of helping researchers to update their understanding of H_2_S donors, function mechanisms, and the pathophysiology of heart diseases. Despite great advances in healthcare, CVDs remain to be the leading cause of deaths worldwide. The role of H_2_S in disease conditions has become one of the research hotspots in recent years. Numerous studies have proved that H_2_S plays a crucial role in cardiovascular homeostasis, so it has potential in the treatment of CVDs. Over the years, researchers have been continuously optimizing H_2_S donors and simultaneously exploring the treatment potential of H_2_S and its associated mechanisms. Generally, H_2_S induces the regulation of cellular activities via S-sulfuration and S-nitrosylation post-translation modifications of various proteins (Table 1). H_2_S has been shown to have a protective effect against variety of CVDs such as CF, CH, and CVC in animal models. Despite the positive results observed following the treatment of various CVDs models with H_2_S donors, the outcome mainly depends on the characteristics of the donor, cell type, and concentration used. For example, treatment with the fast-releasing donor NaHS has been reported to elevate HIF-α levels, while the slow-releasing donor GYY4137 decreases the levels [78,81,105]. Similarly, treatment with NaHS has been shown to sulfurate and increase the activities of c-jun/AP-1 in macrophages [124], while inhibition of the JNK pathway has been reported in cardiomyocytes and epithelial cells A549 [104,125]. Although the drugs have been demonstrated to improve cardiac functioning, it is worth exploring the cause of the observed discrepancy for better drug selectivity. Currently, studies on the effects of H_2_S on CVDs are gradually moving towards research on microRNA and mitochondrial pathways. Despite the many novel results that have been obtained, more complete mechanisms still need to be further clarified. In addition, the effects of H_2_S on CVDs have mainly been detected in cellular and animal model experiments. Whether H_2_S could exert similar effects in humans still needs to be confirmed by more clinical trials. Moreover, the clearance mechanisms of H_2_S-releasing drugs need to be further examined for the development and application of novel drugs in treating CVDs.

Besides, in recent years, nanoparticles have been reported to be efficient in transporting potential drugs to specific targets. For example, single-walled carbon nanotubes have been reported to effectively deliver tyrosine phosphatase inhibitor 1 to macrophages, resulting in macrophage efferocytosis and associated anti-atherosclerotic properties [227]. Nanoparticle-mediated transportation of drugs shows less toxicity to the body and high solubility, specificity, and biodegradability. Therefore, new studies should focus on the use of nanocarriers to deliver potential donors to treat CVDs. With this novel direction, high efficiency and less adverse events can be attained. Lastly, with further research focusing on improving the efficiency of these donors, it is evident that H_2_S-releasing drugs will soon be employed to treat CVDs, especially hypertension and atherosclerosis, due to the progress in these research fields.

## Figures and Tables

**Figure 1 ijms-22-02194-f001:**
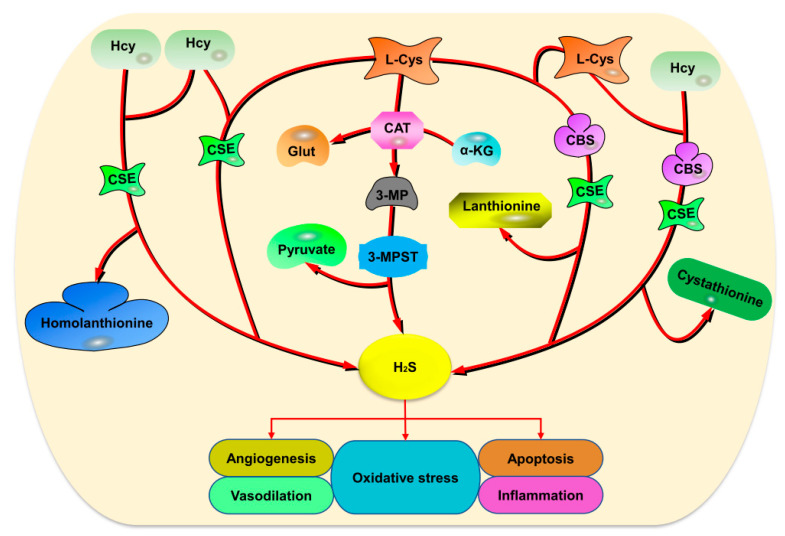
The enzymatic production of H_2_S is catalyzed by CSE, CBS, and 3-MPST. CSE catalyzes the production of H_2_S from the condensation of Hcy or the reaction of l-Cys and Hcy. Furthermore, 3-MPST synthesizes the gas from 3-MP produced from l-Cys by CAT. Additionally, CBS catalyzes the synthesis of H_2_S from two l-Cys molecules or l-Cys and Hcy reaction with the existence of CSE. Abbreviations: H_2_S: Hydrogen sulfide; Hcy: Homocysteine; l-Cys: l-cysteine; CSE: Cystathionine gamma lyase; CBS: Cystathionine beta synthase; 3-MPST: 3-mercaptopyruvate sulfur transferase; Glut: Glutamine; α-KG: Alpha-ketoglutarate; 3-MP: 3-mercaptopyruvate.

**Figure 2 ijms-22-02194-f002:**
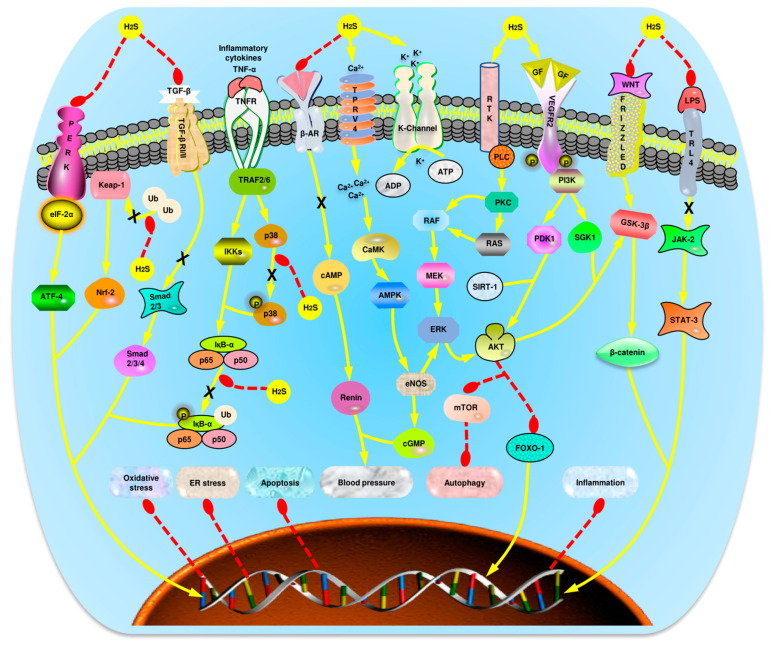
The summary of key pathways regulated by H_2_S donors in the treatment of CVDs. From left to right: H_2_S suppresses the activation of PERK/eIF2α/ATF4 pathway and its subsequent effects on ER stress. Next, H_2_S induces the elevation of Keap-1 by preventing its ubiquitylation, thereby promoting activation of the Keap-1/Nrf-2 pathway and the corresponding antioxidative features. Furthermore, H_2_S suppresses activation of the TGF-β/Smad-2/3 pathway to initiate anti-inflammatory properties. Next, H_2_S inhibits activation of the NF-қB pathway by preventing the phosphorylation of p65 and IқB-α degradation, thereby promoting anti-inflammatory activities. Moreover, H_2_S attenuates the phosphorylation of p38, resulting in inhibition of the pathway and elevation of anti-apoptotic and anti-oxidative properties. Next, H_2_S donors inhibit the activation of β-adrenergic receptors and their downstream effects on the stimulation of cAMP and blood pressure elevation. Additionally, H_2_S activates TPRV4 and KATP channels to mediate angiogenesis and vasodilation, thus regulating blood pressure. By activating TPRV4, H_2_S also activates the AMPK/mTOR pathway, thereby reducing autophagy. Besides, H_2_S activates RTK resulting in the downstream activation of PLC/PKC/Ras/Raf/Mek/ERK pathway and the promotion of anti-apoptotic activities. Furthermore, H_2_S enhances VEGFR-2-mediated activation of the PI3K/AKT/FOXO-1 signaling pathways, resulting in the inhibition of autophagy and apoptotic activities. H_2_S further inhibits activation of the Wnt/β-catenin pathway, resulting in the suppression of pro-apoptotic and ROS activities. Moreover, H_2_S inhibits the JAK-2/STAT-3 pathway, resulting in anti-inflammatory, anti-apoptotic activities, and suppression of ER stress. Abbreviations: H_2_S: Hydrogen sulfide; CVDs: Cardiovascular diseases; PERK: Protein kinase RNA-like endoplasmic reticulum kinase; eIF2α: Eukaryotic initiation factor 2 alpha; ATF-4; activating transcription factor 4; Keap-1: Kelch-like ECH associated protein 1; Ub: Ubiquitin protein; Nrf-2: Nuclear factor erythroid 2-related factor 2; TGF-β: Transforming growth factor beta; Smad: Small mothers against decapentaplegic homologs; TNF-α: Tumor necrosis factor alpha; TNFR: Tumor necrosis factor receptor; TRAF2/6: TNF-receptor associated factor 2/6; IKKs: Kinase of kappa B inhibitors; NF-қB: Nuclear factor kappa B; MAPK: Mitogen activated protein kinase; β-AR: Beta-adrenergic receptors; cAMP: Cyclic adenosine monophosphate; AMPK: AMP-activated protein kinase; CaMK: Calcium/calmodulin-dependent kinase; PLC: Phospholipase C; PKC: Protein kinase C; Raf: Rapidly accelerated fibrosarcoma; eNOS: Endothelial nitric oxide synthase; ERK1/2: Extracellular signal-regulated kinase-1/2; VEGFR-2: Vascular endothelial growth factor receptor-2; PI3K: Phosphoinositide 3-kinase; PDK1: Pyruvate dehydrogenase kinase 1; SGK1: Serum and glucocorticoid-regulated kinase 1; AKT: Protein kinase B; Wnt: Wingless int-1 protein; GSK3-β: Glycogen synthase kinase-3 beta; β-catenin: Beta catenin; mTOR: Mammalian target of rapamycin; FOXO-1: Forkhead box O1; JAK-2/STAT-3: Janus kinase 2/signal transducer and activator of transcription-3.

**Table 1 ijms-22-02194-t001:** The post-translation modification of proteins mediated by H_2_S donors and their corresponding effects.

H_2_S Donors	Cysteine Residues	Resulting Effects	References
NaHS	Cys-269 of c-jun	Induces anti-inflammatory properties	[177]
GYY4137	Cys-664 of specificity protein-1	Reduces myocardial hypertrophy	[170]
NaHS	Cys-44 of muscle ring finger-1	Improves cardiac function	[189]
NaHS and GYY4137	Cys-252, -225, -307, and -310 of CSE	Induces anti-atherosclerotic properties	[195]
NaHS	S-nitrosylation of aortic VSMC proteins	Reduces the progression of atherosclerosis	[196]
GYY4137	Cys-151 of Keap-1	Promotes antioxidative properties	[197]
GYY4137	S-sulfuration of multiple proteins including cathepsin B, fatty-acid binding protein, and glutathione peroxidase 1	Reduces pro-oxidative events	[198]
NaHS and GYY4137	S-sulfuration at the two Zn^2+^ domains of the SIRT-1	Reduces atherosclerotic plaques	[200]

Abbreviations: H_2_S: Hydrogen sulfide; CSE: Cystathionine gamma-lyase; NaHS: Sodium hydrosulfide; Cys: Cysteine; VSMC: Vascular smooth muscle cells; Keap-1: Kelch-like ECH associated protein 1; SIRT-1: Sirtuin-1.

## Data Availability

Not applicable.

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
