# Peer review of "The Potential of Hydrogen Sulfide Donors in Treating Cardiovascular Diseases"

_ijms, 2021, doi:10.3390/ijms22042194_

Round 1

Reviewer 1 Report

Major Comments

  1. The Authors describe this review manuscript as an update on the role of hydrogen sulfide (H2S) in cardiovascular physiology and pathology, with a focus on the cardioprotective effects of H2S donors. While the maunscript comprehensivley reviews the literature on this topic up to the present time, the Authors make little attempt to highlight what are the most recent advances in the field. Out of 212 references, approximately 25 are from 2019 and 9 are dated 2020, and the most current research findings are lost in the extensive reporting of studies over the past 25 years. Thus, this appears to be simply another review in a field where many such reviews exist already, rather than a selective update.

  1. The manuscript needs extensive editing, both in sentence structure and technical writing style. There are many long sentences lacking punctuation, which makes it difficult for the reader to grasp the meaning of the sentence. Here is one example from page 11:

“A recent study by Sikura et al. investigated the effect of H2S donors (NaHS, Na2S, GYY4137, AP67 and AP72) on the calcification development of osteoblastic transdifferentiation of valvular interstitial cells and apolipoprotein E-/- mice fed with atherogenic diet, and found that in cellular model, the donors inhibits osteoblasts transdifferentiation in a dose-dependent manner meanwhile the downregulation of CSE or CBS exacerbates the condition, suggesting that endogenous CSE/CBS-H2S system is involved in calcification of heart valves.”

  1. In addition, there are numerous examples of incorrect English language, which I have tried to list under minor Comments, although this list is far from complete. These language errors make the meaning of many sentences unclear.

Minor Comments

Abstract

Line 26 – There is no such English word as “researches.” It should be “Previous research shows ...”

Introduction

Line 37 – “Targeting the hearts” should be “targeting the heart” (singular)

Line 39 – The sentence “CVD are the world main cause of mortality” is not colloquial English. It would be clearer if written “CVD are the main cause of mortality world-wide.”

Line 42 – Smoking is misspelled as “Smocking.” (throughout)

Line 59 –The meaning is unclear in the sentence “… prior to 1990, most of the studies involving focused on…” Involving what? Re-phrase the sentence.

Line 61 – In “several essential evidences,” the word “evidence” is always singular.

Line 62 – “muscle relaxation” not “relaxations.”

Section 2, Endogenous H2S Production

Line 81 – “regulation [of the] smooth muscle reflex.” (missing words)

Line 86 - “the later helps in activating” should be “the latter.”

Line 92 – What is meant by “Study shows that …”? Which Study?

Line 103 – “despite its pharmacological inhibition to result in over 80% H2S decline” should be “despite its pharmacological inhibition resulting in over 80% decline in H2S [levels?]”

Line 106 – When the Authors say “the enzyme is robust in aortic tissue than coronary artery”, do they mean “the enzyme is more robustly expressed in aortic tissue than in coronary artery”?

Line 110 – What does “3-MPST is also well expressed in vascular system” mean? Do you mean “3-MPST is also strongly expressed in the vascular system”?

Line 128 – “The expressions” should be “The expression.”

Section 3, Exogenous H2S

Line 132 – Do not begin a sentence with the words “Study shows … ”It is meaningless. What “study”?

Line 154 - “there are no available evidences”, should be “there is no available evidence” (singular).

Line 156 – “current researches” should be “current research.”

Line 157 – “their clinical application faces by many challenges” should be “faces many challenges.”

Line 186 – “release the gas up on hydrolysis…” should be “release the gas upon hydrolysis…”

Section 4 Role of H2S in vascular functioning

Line 216 – What is “PCG-1abeam”? I did not find the word beam in reference 69.

Line 250 – “above evidences” should be “evidence.”

Line 275 – as above.

Line 335 – What is “… with the medium activation of autophagy”? Do you mean “With moderate activation of autophagy?”

Line 343 – “Smoking” is misspelled “smocking.”

Section 5 H2S and CVD

Line 378 – “an age dependent dual effects” (mix of singular and plural words).

Line 428 – In “Moreover, study shows…” the meaning is unclear.Line 498 – “smoking” is spelled “smocking.”

Line 512 – What are “current applicable guidelines”? Do you just mean “current guidelines”?

Line 523 – “research” not “researches.”

Line 643 - “one of the main researches focus” should be “research focus.”

Author Response

Dear Editor and Reviewer

Thanks a lot for your comments on the manuscript previously titled “Advances in the treatment of cardiovascular diseases with hydrogen sulfide donors”. We have checked the manuscript carefully and seriously, and revised it point-by-point to all the comments. The replies to the comments are listed below:

Comments and Suggestions for Authors

Reviewer 1

Major Comments

  1. The Authors describe this review manuscript as an update on the role of hydrogen sulfide (H2S) in cardiovascular physiology and pathology, with a focus on the cardioprotective effects of H2S donors. While the maunscript comprehensivley reviews the literature on this topic up to the present time, the Authors make little attempt to highlight what are the most recent advances in the field. Out of 212 references, approximately 25 are from 2019 and 9 are dated 2020, and the most current research findings are lost in the extensive reporting of studies over the past 25 years. Thus, this appears to be simply another review in a field where many such reviews exist already, rather than a selective update.

Thanks for the comment. We have improved the manuscript by adding most recent papers and removing most of the old ones basing on the content and significance to the manuscript.

  1. The manuscript needs extensive editing, both in sentence structure and technical writing style. There are many long sentences lacking punctuation, which makes it difficult for the reader to grasp the meaning of the sentence. Here is one example from page 11: “A recent study by Sikura et al. investigated the effect of H2S donors (NaHS, Na2S, GYY4137, AP67 and AP72) on the calcification development of osteoblastic transdifferentiation of valvular interstitial cells and apolipoprotein E-/- mice fed with atherogenic diet, and found that in cellular model, the donors inhibits osteoblasts transdifferentiation in a dose-dependent manner meanwhile the downregulation of CSE or CBS exacerbates the condition, suggesting that endogenous CSE/CBS-H2S system is involved in calcification of heart valves.”

Thanks for the comment. We have modified the manuscript and edited it thoroughly, and the longer and confusing paragraphs have been revised. A for the above sentence it now reads “A recent study indicates that the H2S donors (NaHS, Na2S, GYY4137, AP67, and AP72) can significantly inhibit osteoblasts transdifferentiation in a dose-dependent manner, meanwhile the downregulation of CSE or CBS exacerbates the condition, suggesting that endogenous CSE/CBS-H2S system is involved in calcification of heart valves [174]”.

  1. In addition, there are numerous examples of incorrect English language, which I have tried to list under minor Comments, although this list is far from complete. These language errors make the meaning of many sentences unclear.

Thanks. We have addressed each one mistake and correct it accordingly.

Minor Comments

Abstract

  1. Line 26 – There is no such English word as “researches.” It should be “Previous research shows ...”

Thanks for the correction. The line now reads “Studies show that H2S plays an important role in cardiovascular homeostasis and in most cases, H2S has been reported to be diminished in cardiovascular diseases (CVD) models.”.

Introduction

  1. Line 37 – “Targeting the hearts” should be “targeting the heart” (singular)

Thanks. The mistake has been corrected and the line now reads “are the non-communicable diseases targeting the heart (rheumatic heart disease, coronary heart disease, and congenital heart disease) and blood vessels (cardiovascular, peripheral arterial, deep vein thrombosis, and pulmonary embolism)”.

  1. Line 39 – The sentence “CVD are the world main cause of mortality” is not colloquial English. It would be clearer if written “CVD are the main cause of mortality world-wide.”

Thanks. We have corrected the mistake and the sentence now reads “According to the 2017 statistics, CVD are the main cause of mortality worldwide, accounted for over 31% of all global death (55.98 million) and over 43% of all non-communicable diseases-caused deaths (41.1 million) [1].”.

  1. Line 42 – Smoking is misspelled as “Smocking.” (throughout)

Thanks. The spelling mistake has been corrected and crosschecked throughout the manuscript.

  1. Line 59 –The meaning is unclear in the sentence “… prior to 1990, most of the studies involving focused on…” Involving what? Re-phrase the sentence.

Thanks for the correction. We have addressed the mistake and updated the information, in the current manuscript it the sentence reads “However, it is important to note that prior to 1990, most of the studies involving H2S focused on its hazardous effects and very few addressed the positive role of the gas in the body”.

  1. Line 61 – In “several essential evidences,” the word “evidence” is always singular.

Thanks. The mistake has been corrected and crosschecked throughout the manuscript.

  1. Line 62 – “muscle relaxation” not “relaxations.”

Thanks. The error has been corrected in the current version.

Section 2, Endogenous H2S Production

  1. Line 81 – “regulation [of the] smooth muscle reflex.” (missing words)

Thanks. The sentence has been corrected and now reads “CSE and 3-MPST have been reported to be highly expressed in vascular tissues and associated with the regulation of the smooth muscle reflex [20, 21]”.

  1. Line 86 - “the later helps in activating” should be “the latter.”

Thanks. The misspelled word has been corrected in the current version.

  1. Line 92 – What is meant by “Study shows that …”? Which Study?

Thanks. The correction has been made throughout the manuscript.

  1. Line 103 – “despite its pharmacological inhibition to result in over 80% H2S decline” should be “despite its pharmacological inhibition resulting in over 80% decline in H2S [levels?]”

Thanks. We have addressed and updated the sentence accordingly, it now reads “Although, a study by Fu et al. could not detect CSE protein in mouse or rat cardiac tissue or cultured rat cardiac myocytes despite its pharmacological inhibition resulting in over 80% decline in H2S levels [33]”.

  1. Line 106 – When the Authors say “the enzyme is robust in aortic tissue than coronary artery”, do they mean “the enzyme is more robustly expressed in aortic tissue than in coronary artery”?

Thanks. The line has been revised and it now reads “Besides, in high-fat diet treated human aortic endothelial cells, the enzyme is more robustly expressed as compared to controls [35]”.

  1. Line 110 – What does “3-MPST is also well expressed in vascular system” mean? Do you mean “3-MPST is also strongly expressed in the vascular system”?

Thanks. The line has been corrected and now reads “3-MPST is also strongly expressed in the vascular system and localized in the vascular endothelium of the thoracic aorta, coronary aorta, and coronary artery [20, 35, 37]”.

  1. Line 128 – “The expressions” should be “The expression.”

Thanks. The mistake has been corrected and crosschecked throughout the manuscript and corrected accordingly.

Section 3, Exogenous H2S

  1. Line 132 – Do not begin a sentence with the words “Study shows … ”It is meaningless. What “study”?

Thanks. The mistake has been corrected and crosschecked throughout the manuscript as for the above line, it now reads “It has been shown that the inhalation of 80 PPM of H2S at 27 or 35 °C for six hours can result in a reversible reduction in cardiovascular function and metabolic rate without affecting the blood pressure in mice [43].”.

  1. Line 154 - “there are no available evidences”, should be “there is no available evidence” (singular).

Thanks. The mistake has been corrected and crosschecked throughout the manuscript.

  1. Line 156 – “current researches” should be “current research.”

Thanks. The mistake has been corrected and crosschecked throughout the manuscript.

  1. Line 157 – “their clinical application faces by many challenges” should be “faces many challenges.”

Thanks. The mistake has been corrected and line now reads “In summary, although these sulfide salts especially NaHS and Na2S are the mainstream of current research, their clinical application faces many challenges therefore their potential is low”.

  1. Line 186 – “release the gas up on hydrolysis…” should be “release the gas upon hydrolysis…”

Thanks. The line has been revised and corrected accordingly.

Section 4 Role of H2S in vascular functioning

  1. Line 216 – What is “PCG-1abeam”? I did not find the word beam in reference 69.

Thanks. We have corrected the mistake and the line now states “A previous study shows that under hypoxia, peroxisome proliferator-activated receptor gamma coactivator 1-alpha can effectively regulate VEGF through the orphan nuclear receptor estrogen-related receptor-α pathway, thereby facilitating the formation of new blood vessels [69]”.

  1. Line 250 – “above evidences” should be “evidence.”

Thanks. The mistake has been corrected and crosschecked throughout the manuscript accordingly.

  1. Line 275 – as above.

Thanks. The mistake has been corrected and crosschecked throughout the manuscript accordingly.

  1. Line 335 – What is “… with the medium activation of autophagy”? Do you mean “With moderate activation of autophagy?”

Thanks. The mistake has been corrected and the sentence now reads “Since moderate activation of autophagy has been shown to have cytoprotective effects [114], it is possible that its hyperactivation results in detrimental effects to the cardiac cells”.

  1. Line 343 – “Smoking” is misspelled “smocking.”

Thanks for the comment. The error has been corrected and crosschecked throughout the manuscript.

Section 5 H2S and CVD

  1. Line 378 – “an age-dependent dual effects” (mix of singular and plural words).

Thanks. The sentence has been revised and it now states “A recent study also reports that knockdown of 3-MPST induces a dual age-dependent effect, where in younger mice it promotes cardioprotection while in adults it is associated with the progression of hypertension [38]”.

  1. Line 428 – In “Moreover, study shows…” the meaning is unclear.

Thanks. Following the revision, the line now reads “With respect to H2S, it has been demonstrated that NaHS treatment can reduce the formation of cardiac fibroblasts by preventing the activation of K+ channels [145]”.

  1. Line 498 – “smoking” is spelled “smocking.”

Thanks for the comment. The error has been corrected and crosschecked throughout the manuscript.

  1. Line 512 – What are “current applicable guidelines”? Do you just mean “current guidelines”?

Thanks. In the current version, the line reads “current guidelines”.

  1. Line 523 – “research” not “researches.”

Thanks. The mistake has been addressed and corrected throughout the manuscript.

  1. Line 643 - “one of the main researches focus” should be “research focus.”

Thanks. The mistake has been addressed accordingly and crosschecked throughout the manuscript.

Reviewer 2 Report

The title of the paper should be modified, since it does not contain randomized human studies, it is review papers containing several experimental observations which have not been used in clinical treatments yet. The advances in the treatment expression in the title means only theoretical approach.

In the exogenous H2S administration section, the inhalation of H2S gas represents only acute experimental treatments, the paragraph should discuss the prolonged treatment as well. 80-300 ppm dose represent a significant amount of gas, and the toxicity of this dose is missing from this part, only one publication reflects severe side effect of the gas administration in an animal experiment.

Although it is review paper, the abbreviations should be introduced in the text in many cases which are missing.  The reviewer and readers need to put a lot of energy into it to figure out their meanings. Although the paper contains an abbreviations section, all of those items must be listed in there.  

In the angiogenesis section the authors claim that H2S and its donors have a positive angiogenetic potential where the VEGF system plays also a positive role, moreover in case of hypoxia H2S represent a positive angiogenetic effect by regulating the HIF-1alpha metabolism. Contrary to this paragraph, at the vascular apoptosis section and the myocardial ischemia/reperfusion injury section, the HIF-1alpha cascade is inhibited by H2S donors. The authors must conclude about this controversial effects of H2S.

The vasodilatation section does not contain well defined data, and comparisons among the biologically active gas molecules. It should contains the H2S, NO and CO effects and their mechanisms.

Again, in the oxidative stress paragraph, there is another inconsistency between the H2S and HIF-1alpha system. A review paper must answer this discrepancy, otherwise the paragraph is pointless.

The grammar of the paper needs a significant improvement, just for an example, instead of plague lesions the correct is plaque lesions.

The two figures need complete new structures, since the H2S is the title of the paper, and activation and inhibition of the H2S should be followed in them. The upper part of the first figure is only a text book quality level, it is not a scientific graph, and moreover the lower part is only a words arranged one after another. The figure 2. again contains well known signal transduction pathways with extremely poor quality.

Author Response

Dear Editor and Reviewer

Thanks a lot for your comments on the manuscript previously titled “Advances in the treatment of cardiovascular diseases with hydrogen sulfide donors”. We have checked the manuscript carefully and seriously, and revised it point-by-point to all the comments. The replies to the comments are listed below:

Reviewer 2

Comments and Suggestions for Authors

  1. The title of the paper should be modified, since it does not contain randomized human studies, it is review papers containing several experimental observations which have not been used in clinical treatments yet. The advances in the treatment expression in the title means only theoretical approach.

Thanks. The title has been changed to “the potential of hydrogen sulfide donors in treating cardiovascular diseases”.

  1. In the exogenous H2S administration section, the inhalation of H2S gas represents only acute experimental treatments, the paragraph should discuss the prolonged treatment as well. 80-300 ppm dose represent a significant amount of gas, and the toxicity of this dose is missing from this part, only one publication reflects severe side effect of the gas administration in an animal experiment.

Thanks. We have revised the paragraph and added the toxicity of the aforementioned dose of H2S gas. It currently states “However, the treatment with high doses of 100-300 ppm H2S in sheep has been associated with the dose-dependent induction of adverse events such as hypoxemia, pulmonary vasoconstriction and systemic vasodilation [46], indicating that high doses of the gas have detrimental effects.”.

  1. Although it is review paper, the abbreviations should be introduced in the text in many cases which are missing. The reviewer and readers need to put a lot of energy into it to figure out their meanings. Although the paper contains an abbreviations section, all of those items must be listed in there.

Thanks. The abbreviations have been introduced accordingly when mentioned for the first time.

  1. In the angiogenesis section the authors claim that H2S and its donors have a positive angiogenetic potential where the VEGF system plays also a positive role, moreover in case of hypoxia H2S represent a positive angiogenetic effect by regulating the HIF-1alpha metabolism. Contrary to this paragraph, at the vascular apoptosis section and the myocardial ischemia/reperfusion injury section, the HIF-1alpha cascade is inhibited by H2S donors. The authors must conclude about this controversial effects of H2

Thanks for the comment. We have addressed the issue and the controversy has now been discussed in the conclusion. “Despite the positive results observed following treatment of various CVD model with H2S donors, the outcome mainly depends on the characteristics of the donor, cell type and concentration used. For example, treatment with a fast-releasing donor NaHS has been reported to elevate HIF-α levels, meanwhile that of slow releasing donor GYY4137 decreases the levels [78, 81, 105]. Similarly, the treatment with NaHS has been shown to sulfurate and increase the activities of c-jun/AP-1 in macrophages [124], while in cardiomyocytes and epithelial cells A549 the inhibition of the JNK pathway have been reported [104, 125]. Although, the drugs have been demonstrated to improve cardiac functioning it’s worth exploring the cause of the observed discrepancy for better drug selectivity”.

  1. The vasodilatation section does not contain well defined data, and comparisons among the biologically active gas molecules. It should contain the H2S, NO and CO effects and their mechanisms.

Thanks. The content of the subsection titled “4.3. Vasodilation

Vasodilation is a vital process involved in the regulation of blood pressure and its dysregulation is considered as a risk factor for CVD [89]. NO, angiotensin II (Ang II), K+, and Ca2+ channels play crucial roles in the regulation of blood flow and vasodilation [14, 90, 91]. The reduction of endogenous H2S-induced vasodilation has been reported in hypertension patients as compared to adults [92]. In mesenteric stem cells, the downregulation of CBS reduces the production of H2S and dilation of mesenteric arteries [93]. Also, treatment with CSE inhibitor DL-propargylglycine enhances vascular resistance and increases bloop pressure in rats, indicating that H2S is involved in the regulation of vasodilation [94]. Correspondingly, the stimulation of H2S production with PLP enzyme significantly elevates the H2S levels and improves vascular relaxation together with oxidative and nitrosative statuses in adult rats [95]. Similarly, treatment with NaHS has also been shown to induce the relaxation of human uterine artery partly by triggering the large conductance calcium-activated and voltage-dependent K+ (BK) channels [96]. An isothiocyanate vasodilating compound, sulforaphane has also been reported to induce its effect by utilizing the CBS/CSE/H2S signaling, resulting in the activation of KATP and BK channels [97]. In addition, H2S also induces vasodilation by promoting NO release and ultimately activating the NO/cGMP/soluble guanylyl cyclase/protein kinase G pathway in pial arteries [98]. A previous study further reveals that treatment with Na2S improves the vasodilation of endothelial cells by activating transient receptor potential cation channel V- 4 [99]. Otherwise, a glucagon-like peptide 1 inhibitor exenatide has been identified to facilitate vasodilation and decrease aortic blood pressure by stimulating H2S, NO, and CO production, and consequently activating the voltage-dependent K+ channel subfamily KQT member 5 type pathways [100]. However, compared to other gasotransmitters, H2S induced a stronger effect. Collectively, these data signify that H2S treatment improves vascular function including vasodilation by regulating ion channels and multiple signaling pathways”.

  1. Again, in the oxidative stress paragraph, there is another inconsistency between the H2S and HIF-1alpha system. A review paper must answer this discrepancy, otherwise the paragraph is pointless.

Thanks. As stated before in comment number 4, the controversy has been discussed in the conclusion.

  1. The grammar of the paper needs a significant improvement, just for an example, instead of plague lesions the correct is plaque lesions.

Thanks. We have crosschecked the manuscript and corrected grammar mistakes accordingly.

  1. The two figures need complete new structures, since the H2S is the title of the paper, and activation and inhibition of the H2S should be followed in them. The upper part of the first figure is only a text book quality level, it is not a scientific graph, and moreover the lower part is only a words arranged one after another. The figure 2. again contains well known signal transduction pathways with extremely poor quality.

Thanks. Concerning the figure 1, the aim of the figure was to summarize the key mechanisms involved in endogenous production of H2S and the mechanisms regulated by the gas, in the current version the quality of the figure has been improved. As for figure 2, we have revised it accordingly. In addition, the quality of this figure has been improved too.

Round 2

Reviewer 1 Report

The revised manuscript is significantly improved in structure and writing style, and the authors have updated the literature covered by this review.  However, the summary sentences at the end of each section simply repeat the Authors’ premise that hydrogen sulfide donors have potential for treating whichever cardiovascular disorder under discussion in that section. With the updated review I would have expected the Authors to highlight where the field is currently in regard to the potential of H2S donors as a therapy for cardiovascular disorders and present a more critical interpretation of the evidence. With hydrogen sulfide being such a multi-functional signalling molecule (as the manuscript describes in great detail), how would H2S donors be administered as a therapy and how could off-target effects be avoided? In which cardiovascular pathology do they see H2S donors having the most therapeutic potential in the near future?  While no Review can be exhaustive of all current publications, some papers that provide conflicting views are not discussed, such as a paper published in IJMS last year that questions whether CSE-derived generation of H2S is obligatory in the cardiac response to myocardial infarction (Ellmers et al., Int. J. Mol. Sci.2020, 21, 4284; doi:10.3390/ijms21124284).

I noticed another minor spelling error in Table 1 – in the final line – “atherosclerotic plaques” is spelled “plagues.”

Author Response

Thanks for the comment. We have improved the summary sentences in each subpart discussing the diseases by providing more details on the interpretation of the reported data and the gap. Also, we have improved the conclusion of the manuscript by adding the following information “Currently, studies on the effects of H2S on CVD are gradually moving towards the research on microRNA and mitochondrial pathways. Despite, many novel results that have been obtained, more complete mechanisms are still needed to be further clarified. In addition, the effects of H2S on CVD are mainly detected in cellular and animal model experiments. Whether H2S could exert similar effects in human still needs to be confirmed by more clinical trials. Moreover, the clearance mechanisms of H2S-releasing drugs need to be further examined for the development and application of novel drugs in treating CVD.

Besides, in recent years nanoparticles have been reported to be efficient in transporting potential drugs to the specific targets. For example, single-walled carbon nanotubes have been reported to effectively deliver tyrosine phosphatase inhibitor 1 to the macrophages, resulting in macrophage efferocytosis and associated anti-atherosclerotic properties [227]. Nanoparticles-mediated transportation of drugs show less toxicity to the body, high solubility, specificity, and biodegradability. Therefore, new studies should focus on the use of nanocarriers to deliver potential donors to treat CVD. With this novel direction, high efficiency and less adverse events can be attained. Lastly, with further research focusing on improving the efficiency of these donors, it is evident that H2S-releasing drugs will soon be employed to treat CVD especially hypertension and atherosclerosis due to the progress in these research fields”.

In addition, the details of the CSE controversy have been added on myocardial ischemia reperfusion injury, and it reads as follows “A previous study shows that the treatment of M-I/R rat with a CSE inhibitor PAG significantly increases infarct size, suggesting the suppression of the gene aggravates the condition [212]. In a recent study, endothelia cell-specific CSE overexpressed transgenic mice have also been revealed to display high protective effects induced through the activation of eNOS and elevation of NO production when subjected to M-I/R [213]. In addition, CSE knockout mice exhibit high endothelial dysfunction but without aggravating the induced M-I/R. Similarly, another recent study indicates that the deletion of CSE in mice increases mean arterial pressure and miR21a expression, however no intensification of M-I/R effects could be observed [214]. These findings suggest that CSE/H2S play crucial role in cardioprotection although the suppression of CSE alone cannot promote the progression of M-I/R”.

Reviewer 2 Report

The authors have accepted the advices and changes have been made according to my criticisms. The important parts of this review are the figures. The modification of the second figure was suggested in order to focus on the H2S effects on cellular signal transductions and cell metabolism. The first two systems in the second figure are the endoplasmic reticulum stress mediator pathway and the Keep-1-Nrf-2 oxidative stress sensor system. The authors include the apoptosis instead of endoplasmic reticulum stress in the figure legends, and writes the H2S inactivates this pathway. If it correct, H2S behaves as an endoplasmic reticulum stress inhibitor. Although this pathway represents the first pathway in the graph, the main text does not discuss the relationship between H2S donors and endoplasmic reticulum stress in details based on original publications. The second pathway is the Keep-1 system. The authors inform us that the H2S activates this antioxidant system, without detailed overview in the main text. Moreover, I am not able to distinguish the inhibitory and activation capability of H2S in the figure two. The figure two needs significant modification. It should have been done after my first review and suggestion.

Author Response

Thanks for the comment. We have modified figure 2 and added more details. In addition, the legend has also been improved and it now reads as follows “Figure 2. The summary of key pathways regulated by H2S donors in the treatment of CVD. From left to right: H2S suppresses the activation of PERK/eIF2α/ATF4 pathway and its subsequent effects on ER stress. Next, H2S induces the elevation of Keap-1 by preventing its ubiquitylation thereby promoting the activation of Keap-1/Nrf-2 pathway and the corresponding anti-oxidative features. Also, the H2S suppresses the activation of TGF-β/Smad-2/3 pathway to initiate anti-inflammatory properties. Next, H2S inhibit the activation of NF-Ò›B pathway by preventing the phosphorylation of p65 and IÒ›B-α degradation thereby promoting anti-inflammatory activities. Moreover, H2S attenuates the phosphorylation of p38 resulting in the inhibition of the pathway and elevation of anti-apoptotic and anti-oxidative properties. Next, H2S donors inhibit the activation of β-adrenergic receptors and their downstream effects in the stimulation of cAMP and blood pressure elevation. Next, H2S activates TPRV4 and KATP channels to mediate angiogenesis and vasodilation, thus regulating blood pressure. By activating TPRV4, H2S also activates AMPK/mTOR pathway thereby reducing autophagy. Besides, H2S activates RTK resulting in the downstream activation of PLC/PKC/Ras/Raf/Mek/ERK pathway and the promotion of anti-apoptotic activities. Also, H2S enhances VEGFR-2-mediated activation of PI3K/AKT/FOXO-1 signaling pathways resulting in the inhibition of autophagy and apoptotic activities. H2S further inhibits the activation of Wnt/β-catenin pathway resulting in the suppression of pro-apoptotic and ROS activities. Moreover, H2S inhibits the JAK-2/STAT-3 pathway resulting in anti-inflammatory, anti-apoptotic activities and suppression of ER stress. Abbreviation: H2S: Hydrogen sulfide; CVD: Cardiovascular diseases; PERK: Protein kinase RNA-like endoplasmic reticulum kinase; eIF2α: Eukaryotic initiation factor 2 alpha; ATF-4; activating transcription factor 4; Keap-1: Kelch-like ECH associated protein 1; Ub: Ubiquitin protein; Nrf-2: Nuclear factor erythroid 2-related factor 2; TGF-β: Transforming growth factor beta; Smad: Small mothers against decapentaplegic homologs; TNF-α: Tumor necrosis factor alpha; TNFR: Tumor necrosis factor receptor; TRAF2/6: TNF-receptor associated factor 2/6; IKKs: Kinase of kappa B inhibitors; NF-Ò›B: Nuclear factor kappa B; MAPK: Mitogen activated protein kinase; β-AR: Beta-adrenergic receptors; cAMP: Cyclic adenosine monophosphate; AMPK: AMP-activated protein kinase; CaMKK: Calcium/calmodulin-dependent kinase kinases; PLC: Phospholipase C; PKC: Protein kinase C; Raf: Rapidly accelerated fibrosarcoma; eNOS: Endothelial nitric oxide synthase; ERK1/2: Extracellular signal-regulated kinase-1/2; VEGFR-2: Vascular endothelial growth factor receptor-2; PI3K: Phosphoinositide 3-kinase; PDK1: Pyruvate dehydrogenase kinase 1; SGK1: Serum and glucocorticoid-regulated kinase 1; AKT: Protein kinase B; Wnt: Wingless int-1 protein; GSK3-β: Glycogen synthase kinase-3 beta; β-catenin: Beta catenin; mTOR: Mammalian target of rapamycin; FOXO-1: Forkhead box O1; JAK-2/STAT-3: Janus kinase 2/signal transducer and activator of transcription-3”.